# Characterization of the Bacterial Biofilm Communities Present in Reverse-Osmosis Water Systems for Haemodialysis

**DOI:** 10.3390/microorganisms8091418

**Published:** 2020-09-15

**Authors:** Juan-Pablo Cuevas, Ruben Moraga, Kimberly Sánchez-Alonzo, Cristian Valenzuela, Paulina Aguayo, Carlos T. Smith, Apolinaria García, Ítalo Fernandez, Víctor L Campos

**Affiliations:** 1Environmental Microbiology Laboratory, Department of Microbiology, Faculty of Biological Sciences, University of Concepción, Concepción 4070386, Chile; juancuevas@udec.cl (J.-P.C.); crvalenz@udec.cl (C.V.); itfernan@udec.cl (Í.F.); 2Microbiology Laboratory, Faculty of Renewable Natural Resources, Arturo Prat University, Iquique 1100000, Chile; rmoraga@unap.cl; 3Laboratory of Bacterial Pathogenicity, Department of Microbiology, Faculty of Biological Sciences, Universidad de Concepcion, Concepcion 4070386, Chile; kimsanchez@udec.cl (K.S.-A.); csmith@udec.cl (C.T.S.); apgarcia@udec.cl (A.G.); 4Faculty of Environmental Sciences, EULA-Chile, Universidad de Concepcion, Concepcion 4070386, Chile; paulinaaguayo@udec.cl; 5Institute of Natural Resources, Faculty of Veterinary medicine and agronomy, Universidad de Las Américas, Sede Concepcion, Chacabuco 539, Concepcion 3349001, Chile

**Keywords:** biofilm/biofouling, reverse osmosis membranes, haemodialysis, metagenomics

## Abstract

Biofilm in reverse osmosis (RO) membranes is a common problem in water treatment at haemodialysis facilities. Bacteria adhere and proliferate on RO membranes, forming biofilms, obstructing and damaging the membranes and allowing the transfer of bacteria and/or cellular components potentially harmful to the health of haemodialysis patients. Our aim was to characterize the bacterial community associated to biofilm of RO membranes and to identify potentially pathogenic bacteria present in the haemodialysis systems of two dialysis centres in Chile. The diversity of the bacterial communities present on RO membranes and potable and osmosed water samples was evaluated using Illumina sequencing. Additionally, bacteria from potable water, osmosed water and RO membrane samples were isolated, characterized and identified by Sanger’s sequencing. The molecular analyses of metagenomics showed that the *phyla* having a greater relative abundance in both dialysis centres were *Proteobacteria* and *Planctomycetes. Pseudomonas, Stenotrophomonas*, *Agrobacterium*, *Pigmentiphaga*, *Ralstonia*, *Arthrobacter*, *Bacteroides* and *Staphylococcus* were bacterial genera isolated from the different samples obtained at both haemodialysis centres. *Pseudomonas* spp. was a bacterial genus with greater frequency in all samples. *Pseudomonas* and *Staphylococcus* showed higher levels of resistance to the antibiotics tested. Results demonstrated the presence of potentially pathogenic bacteria, showing resistance to antimicrobials on RO membranes and in osmosed water in both dialysis centres studied.

## 1. Introduction

Since the incidence of chronic renal failure is increasing around the world, the number of individuals requiring treatment for end-stage renal disease is also rising [1]. Haemodialysis (HD) is the main treatment for patients with severe renal failure. Patients undergoing HD are subjected to large volumes of dialysis fluid (approximately 120 L) in a single dialysis treatment [2].

Considering the repeated large volumes of water each patient is subjected to, ensuring the necessary quality of dialysate solution is a vital aspect for a safe HD because chemical, bacterial and associated endotoxin and bacterial debris contamination can threaten the health of a patient requiring this treatment [3]. Moreover, dialysis patients often have additional comorbidities (e.g., diabetes, hypertension and cardiovascular disease) making them more vulnerable to adverse outcomes [4]. Therefore, the microbiological quality of water present in the accumulation tanks at the dialysis centres is of utmost concern. The Association for the Advancement of Medical Instrumentation (AAMI) recommendations for conventional dialysis water includes its microbiological quality, requiring <100 CFU mL^−1^ total heterotrophic bacteria viable count and <0.25 UE mL^−1^ for endotoxins [5]. In Chile, similar standards are required for HD water, i.e., <100 CFU mL^−1^ total heterotrophic bacteria viable count and <0.03 UE mL^−1^ for endotoxins [5].

Dialysis centres have water accumulation tanks that receive potable water (PW) and supply it to the water pre-treatment system. PW, after passing through physicochemical processes, is subjected to treatment through reverse osmosis (RO). This system uses spirally wounded Thin Film Composite (TFC) membranes, a type of membrane commonly used for water reconditioning. RO membranes are highly efficient, removing approximately 99% of the bacteria present in water treatment systems [6]. Nevertheless, biofouling and flow reduction, caused by microorganisms capable to form biofilms on them, are well-known disadvantages of RO membranes causing their malfunction [7,8] and allowing the transfer of bacteria and endotoxins to dialysed water, a possible hazard for the health of patients undergoing HD [9,10].

During biofilm formation, free-swimming (planktonic) bacteria become firmly attached to surfaces, grow and recruit more planktonic cells [11]. Microbial biofilms contain bacterial cells and extracellular polymeric substances including polysaccharides, proteins and nucleic acids. This matrix favours the adhesion of microorganisms and protects them from various stresses and severe environmental conditions. Biofilms present on RO membranes are multi-genera and they are often thicker and more resistant to antibiotics and other antimicrobial compounds than biofilms formed by only one bacterial genus [2,4,9,11]. Besides being important in microbial ecology, bacterial adhesion is also important in biotechnology, biofouling and wastewater treatment. The ability of bacteria to interact with surfaces depends on particular molecules or structures present on their outer surface. These may be outer membrane proteins or lipopolysaccharides, proteinaceous appendages (fimbriae or flagella) or complex carbohydrates of extracellular capsules [4].

Biofilm of RO membranes have not only an economic negative impact for dialysis centres but also for the health of HD patients. The presence of multi-drug resident bacteria and persistent bacteria in biofilm has been poorly studied, and considering the risk they signify to patients, the aim of the present work was to characterize the bacterial communities associated to the water and RO membranes from two haemodialysis treatment centres and to identify potentially pathogenic bacteria and to determine their resistance to antimicrobials.

## 2. Materials and Methods

### 2.1. Sampling

Samples were collected at two dialysis centres located in the north-central area (DCN) and south-central area (DCS) of Chile. Both centres are located approximately 500 km apart. Samples, 1 L each by triplicate, of potable water (PW) and of osmosed water (OW) stored in the accumulation tanks were collected. Further processing of samples was also processed by triplicate. Reverse osmosis (RO) membranes were also collected. Water samples and RO membranes were transported, at 4 °C, to the Laboratory of Environmental Microbiology, Faculty of Biological Sciences, University of Concepcion, Concepcion, Chile, and stored at that same temperature until further analysis. In order to determine the total and heterotrophic bacterial counts from the RO membranes, two samples of 10 cm^2^ each were aseptically obtained in a laminar flow hood (ZHJH-C 1109C, Zhi cheng, Shangai, Korea) from their initial, middle and final sections, hereon referred as membrane segments A, B or C, respectively. Furthermore, sets of RO membrane segments were obtained for microscopic analysis.

### 2.2. Scanning Electron Microscopy

To study the segments of RO membranes under scanning electron microscopy (SEM), they were washed three times using sterile saline solution and fixed with 2.5% glutaraldehyde. After dehydration, the samples were critical point dried using CO_2_ and coated with gold (S150 Sputter Coater, Edwards/Kiese, Germany). Samples were analysed using a JEOL JSM 6380LV SEM (JEOL, Tokyo, Japan). 

### 2.3. Total and Heterotrophic Bacterial Counts 

To quantify the total and heterotrophic bacterial counts present in RO membranes, it was necessary to previously sonicate segments of RO membranes in 100 mL phosphate-buffered saline (PBS) (pH 7.0) in an ultrasonic bath (Elma/S30, Singen, Germany) adjusted at 100 W for 2 min.

To determine total bacterial counts, 10 mL samples (from PW, OW or previously sonicated RO segments) were fixed in formaldehyde solution (4% *w/v*) and filtered using white polycarbonate membranes (0.2 μm, Ø47 mm). DAPI (4,6-diamino-2-phenylin doldihydrochloride dilactate) staining was added to a final concentration of 1 mg mL^-1^. Bacterial cells were counted using a Motic BS-310 epifluorescence microscope equipped with a MotiCam Pro 285A digital camera (Motic, Richmond, Canada). 

To count heterotrophic bacteria, 100 mL of samples (from PW, OW or previously sonicated RO segments) were place on Plate Count Agar (PCA) (Merck, NJ, USA) and incubated at 37 °C. The results were read at 44 ± 4 h, as recommended by the Standing Committee of Analysts, United Kingdom guidelines (Environment Agency 2012).

### 2.4. Polymerase Chain Reaction (PCR)-Denaturing Gradient Gel Electrophoresis (DGGE) and Analysis of DGGE Profiles

Total DNA from RO membrane segments and from water (PW and OW) was extracted using the UltraClean soil DNA extraction kit (MO BIO Laboratories Inc., Carlsbad, CA, USA) following the protocol provided by the manufacturer. Total DNA of each sample was amplified using 16S rRNA universal primers EUB 9–27 and EUB 1542 (Guzmán-Fierro et al. 2015) [12]. Nested PCR was performed using the primer pair 341f and 534r with a GC clamp (CGCCC GCCGC GCGCG GCGGG CGGGG CGGGG GCACG GG GGG) according to Mafla et al. [13]. DGGE was done using a DGGE 1001 system (C.B.S. Scientific Company Inc., San Diego, USA) according to Campos et al. [14].

For DGGE profile analyses, DGGE gels were digitized using a photo-documentation system MaestroGen (MaestroGen Inc., Hsinchu, Taiwan). The bands were scored as present or absent at each position in the gel, using the Gel-Pro Analyzer 4.0 software package (Media Cybernetics, Silver Spring, MD, USA). For the analysis of banding profiles, a binary matrix was constructed based on the presence (1) or absence (0) of individual bands in each lane. A multidimensional scaling diagram (MDS) was constructed using the Bray Curtis algorithm, according to Guzman-Fierro et al. (2015) [12]. 

### 2.5. Genomic DNA Extraction and 16S rRNA Gene Massive Sequencing

Genomic DNA was extracted directly from PW and OW samples and from the three segments of RO membranes of both dialysis centres using the PowerSoil DNA kit (MoBio, Carlsbad, CA, USA) according to the manufacturer’s instructions. The DNAs extracted were subsequently purified using the UltraClean 15 DNA Purification Kit (MoBio, Carlsbad, CA, USA). The quality and concentration of DNAs were checked by UV/Vis spectroscopy (NanoDrop ND-1000, Peq-lab, Erlangen, Germany).

Total DNA extracted from RO membranes (combination of the three segments) and PW samples of the two-dialysis centres were quantified and sequenced. The DNA extracted from OW samples was not sufficient to perform the assays. Illumina sequencing was performed at Genoma Mayor, Universidad Mayor, Santiago, Chile. The raw data was analysed using the Mothur bioinformatics analysis software (version 1.35.1; https://mothur.org) using, unless indicated, default options. Reads under 200 bp were discarded. Denoising of reads was achieved using the Mothur’s platform “pre.cluster” command to remove sequences probably caused by errors and assemble reads differing only by 2bp. Chimeric sequences were identified and removed using the UCHIME algorithm [15]. The remaining sequences were classified using the SILVA database.

### 2.6. Bacterial Isolation

PW and OW samples were concentrated using 0.22 μm cellulose nitrate filters (Corning Inc., NY, USA). Then the filters were sonicated in a bath (Elmasonic P 120 H ultrasonic, Singen, Germany) containing 10 mL PBS, pH 7.0. Segments of RO membranes were also sonicated in the same buffer mentioned above. For isolation of bacterial strains, 100 µL of each sonicated filter or RO membrane segment were disseminated on the surface of Petri dishes containing BBL nutritive agar (Becton-Dickinson and Company, NJ, USA). Plates were incubated at 30 °C for 24–48 h under aerobic conditions. The selected colonies were stored at −80 °C for further analysis. 

### 2.7. Identification of Heterotrophic Aerobic Bacteria

DNA was extracted from the bacterial isolates using the InstaGene Matrix kit (BIO-RAD, Hercules, CA, USA), according to the manufacturer’s indications. Bacterial DNA of each isolate was amplified by PCR according to a method described by Wang et al. (2008) using 16S rDNA universal primers EUB 9-27f and EUB 1542r (5′-GAGTTTGATCCTGGCTCAG-3′) and (5′-AGAAAGGAG GTGATCCAGCC-3′). PCR products were sequenced by the Sanger’s method using an ABI PRISM 3500 xL Genetic Analizer (Applied Biosystems, Foster City, CA, USA). The sequences were analysed by means of Basic Local Alignment Search Tools (BLAST).

### 2.8. Antimicrobial Susceptibility Assay 

Antimicrobial susceptibility tests were performed to the isolated strains obtained from water samples and RO membrane samples. The standard disk diffusion technique was carried out according to the recommendations of the Clinical Laboratory Standards Institute (National Committee for Clinical Laboratory Standards 2017). Isolated strains were incubated in 5 mL Tryptic Soy Broth (Becton Dickinson, Franklin Lakes, NJ, USA) for 18 h at 30 °C, serially diluted until reaching a turbidity similar to 0.5 McFarland. Each strain was disseminated in Petri dishes containing Müller–Hinton agar (Merck, Darmstadt, Germany). Neo-Sensitabs disks (Rosco, Taastrup, Denmark) used were ampicillin (10 mg), chloramphenicol (30 mg), tetracycline (30 mg), cefotaxime (30 mg), ciprofloxacin (5 mg), trimethoprim/sulfamethoxazole (1.25 mg/23.75 mg) and gentamicin (10 mg). Petri dishes were incubated overnight at 30 °C. Qualitative antimicrobial susceptibility patterns (S: Susceptible, I: Intermediate, or R: Resistant) were determined after measuring the diameters of inhibition halos. *E. coli* ATCC 25922 strain and *P. aeruginosa* ATCC 27853 strain were used as control.

### 2.9. Statistical Analysis

Clustering analysis and MDS were performed using the PRIMER V.6 software package (Plymouth Routines in Multivariate Ecology Research, Auckland, New Zealand). Statistical significance of variance in indexes was evaluated using a one-way ANOVA using GraphPad Prism 5 software (GraphPad Software, San Diego, CA, USA). *p* values equal to or less than 0.05 were considered as significant. Statistical significance of variance in indexes was evaluated with a one-way analysis of variance (ANOVA).

## 3. Results

### 3.1. Scanning Electron Microscopy (SEM) Observation of RO Membranes

SEM observations of the three segments of RO membranes for both haemodialysis centres showed the presence of bacterial biofilms on their surface, revealing the presence of thin fibrillar pili at the bacterial cell surfaces in the initial and final segments (A and C, respectively) (Figure 1 and Figure 2). Furthermore, it was observed that the biofilm of the membranes of both centres was composed of a heterogeneous microbial community.

### 3.2. Total Cell and Viable Heterotrophic Bacterial Counts

Total cell numbers were determined by DAPI staining. Regarding the total cell counts of the RO membranes of both centres, the results for the DCN centre were 4.1 × 10^7^, 2.7 × 10^7^ and 5.3 × 10^7^ cells mL^−1^ for the segments DCN A, DCN B and DCN C, respectively. Similar counts were found in the southern centre, with figures of 2.1 × 10^7^, 3.2 × 10^7^ and 1.7 × 10^7^ cells mL^-1^ for the segments DCS A, DCS B and DCS C, respectively. No significant differences were detected between the segments of the RO membranes (*p* > 0.05) from both dialysis centres. The heterotrophic bacterial counts of RO membrane segments DCN A, DCN B and DCN C were 2.1 × 10^4^, 2.4 × 10^4^ and 1.7 × 10^4^ CFU mL^−1^, respectively. Similar counts were obtained from the RO membrane segments DCS A, DCS B and DCS C with figures of 3.3 × 10^4^ and 3.8 × 10^4^ and 1.2 × 10^4^ CFU mL^−1^, respectively.

The results for the total bacterial counts in water samples were 6.4 × 10^2^ and 7.3 × 10^3^ cells mL^−1^ for DCN PW and DCN OW, respectively and 3.7 × 10^2^ and 5.7 × 10^3^ cells mL^-1^ for DCS PW and DCS OW. Furthermore, there was no significant difference between PW and OW (*p* > 0.05) samples. The results for the heterotrophic bacterial counts in water samples were 6 x 10^1^ CFU mL^-1^ for DCN PW and 7.1 × 10^2^ CFU mL^−1^ for DCN OW and 3 × 10^1^ CFU mL^−1^ for DCS PW and 4.1 × 10^2^ CFU mL^−1^ for DCS OW. No significant differences were detected when comparing the segments of the RO membranes from both dialysis centres (*p* > 0.05). Similarly, no significant differences were observed between PW and OW (*p* > 0.05) of both centres.

### 3.3. Analysis of Similarity of Bacterial Communities by DGGE

DGGE profile bands or operational taxonomic units (OTUs) were analysed using the Bray–Curtis correlation. A distance matrix was calculated, and a cluster analysis was performed which resulted in a multidimensional scaling (MDS). The MDS analysis of the banding patterns obtained by DGGE revealed that the sampled segments of the RO membranes showed a high degree of similarity between them, 98 and 95% for DCN and DCS, respectively (Figure 3 and Figure 4). Analysis of similarities (ANOSIM) analysis (R = 0.004, P = 0.3140/R = 0.008, P = 0.4540) showed that differences between abundance of OTUs were not significant. These results were consistent with the hierarchical cluster analysis (Bray–Curtis index), which clearly indicated the high similarity between the three segments of RO membrane of both dialysis centres. In addition, MDS analysis revealed that the sampled segments of the RO membranes showed a low degree of similarity, 60 and 65% with DCN-PW and DCS-PW, respectively (Figure 3 and Figure 4). ANOSIM analysis (R = 0.41, P=0.001/R = 0.23, P=0.001) determined significant differences between samples in terms of abundance of OTUs, for both dialysis centres.

### 3.4. Analysis of the Bacterial Communities’ Composition 

The metagenomics analysis of the total DNA extracted from the segment of RO membranes and PW samples showed the relative abundances of the main *phyla*. The main phyla in the DCN RO membrane were *Proteobacteria* (49%), *Planctomycetes* (26%) and *Acidobacteria* (9%), while in the DCS RO membrane they were *Proteobacteria* (86%), *Actinobacteria* (5%) and *Planctomycetes* (3%). Regarding potable water, the main phyla in DCN PW were *Proteobacteria* (73%), *Planctomycetes* (12%) and *Bacteroidetes* (5%) and in DCS PW they were *Proteobacteria* (99.872%), *Firmicutes* (0.0401%) and *Bacteroidetes* (0.0327%) (Figure 5). 

A *Proteobacteria* class differentiation analysis was performed. With respect to the membranes, the relative abundances in DCN RO were: α-*proteobacteria* (31%), *β-proteobacteria* (1%), *γ-proteobacteria* (1%), *δ-proteobacteria* (12%) and non-classified *Proteobacteria* (4%), while in the DCS RO membrane, they were *α-proteobacteria* (9%), *β-proteobacteria* (68%), *γ-proteobacteria* (7%) and *δ-proteobacteria* (2%). The results for potable water in DCN PW were *α-proteobacteria* (36%), *β-proteobacteria* (18%), *γ-proteobacteria* (6%), *δ-proteobacteria* (11%) and non-classified *Proteobacteria* (2%), while in DCS PW, they were *α-proteobacteria* (42%), *γ-proteobacteria* (55%) and non-classified *Proteobacteria* (3%) (Figure 5).

### 3.5. Identification of Isolates

A total of 29 morphologically different isolates were initially obtained from both dialysis centres (15 and 14 morphotypes for DCN and DNS, respectively). Subsequent microscopy, biochemical profiles and 16 s rRNA analyses suggested that some of these isolates likely belonged to the same taxa. Finally, 15 different isolates were identified. PCR products of 16s rDNA were sequenced by the Sanger’s method and processed by BLAST with 96%–98% identity. The closest GenBank matches of 16S rDNA sequences revealed the presence of genera *Pseudomonas, Stenotrophomonas, Agrobacterium, Pigmentiphaga, Ralstonia, Arthrobacter* and *Bacteroides* in DCN and *Pseudomonas, Stenotrophomonas* and *Staphylococcus* in DCS (Table 1).

### 3.6. Antimicrobial Susceptibility Analysis

Antimicrobial susceptibility tests were applied to strains isolated from RO membranes and water samples. The strains showed a broad resistance to antibiotics, it being higher for ampicillin, chloramphenicol and cefotaxime followed by trimethoprim/sulfamethoxazole, gentamicin, tetracycline and ciprofloxacin (Table 2). In addition, most of the isolates showed intermediate sensitivity to at least one of the antibiotics tested. The results showed that the bacterial strains belonging to the genera *Pseudomonas* and *Staphylococcus* showed higher levels of resistance to the antibiotics tested (Table 3).

## 4. Discussion

One of the important complications in patients receiving HD is bloodstream infection. HD patients have 26 times higher risk of bacteraemia than the general population, increasing the risk of systemic infections, hospitalization and patient death. The Centres for Disease Control and Prevention (CDC) and the American Society of Nephrology (Transforming Nephrology Dialysis Safety Initiative) have focused on finding methodologies for eliminating bloodstream infections in HD patients [16,17], indicating that there is a need to generate prophylactic measures to allow the reduction of infections in these patients.

The use of catheters is one of the main risk factors for acquiring infections in HD patients [16]. Since HD patients expose their circulatory system to approximately 300–400 L of water per week, the quality of the water used for this purpose plays an important assure a safe procedure [4]. Although dialysis centres have detailed protocols for the analysis of water to be used, its contamination with pathogenic microorganisms can occur due to the use of inappropriate water sources, to failures in the maintenance or disinfection of HD equipment and to problems in the water treatment system, such as the malfunction or RO membranes used to purify water [18].

RO is one of the most widely used techniques for treating water to make it suitable for HD. However, sometimes biofilms may develop on the membranes used in the purification process, affecting water treatment due to the decrease in the effectiveness of the permeate [19,20]. In this study, a microbiological analysis of water and RO membranes used for water purification for HD was done in two dialysis centres located in the north-central area (DCN) and south-central area (DCS) of Chile.

SEM of the RO membranes of both centres demonstrated the presence of a heterogeneous biofilm, similar to those described by other authors [21,22,23]. SEM analyses showed fibrillar structures (pili) similar to the curli structures described by other authors [24] in all the segments of RO membranes. These structures play an important role to sustain the biofilm, firmly anchoring bacteria, providing a remarkable stability to the biofilm to withstand prolonged strong hydrodynamic conditions without disruption.

The analysis of the bacterial community by a DGGE profile suggested that there were no differences in bacterial composition between the segments of the membranes of both centres; therefore, metagenomic analyses were performed by mixing total DNA extracted from the three segments of each RO membrane and from PW samples. *Proteobacteria* phylum was the predominant biofilm phylogenetic group in the RO membranes of both dialysis centres, being ubiquitously distributed in freshwater [3,23]. If the high relative abundance of *Proteobacteria* is compared to the strains isolated from the RO membranes it can be concluded that most biofilm forming *Proteobacteria* present on the membranes are non-cultivable or dormant bacteria. It must be also considered that the samples of RO membranes were obtained from RO membranes at the end of their useful life, largely exceeding six months of use, and the different capability of PW bacteria to form biofilms may considerably affect the proportion of particular bacterial species in the mature biofilm. Thus, the bacterial community present in a biofilm does not necessarily must replicate the bacterial diversity of the water. This is supported by the fact that chlorine and oligotrophic conditions can induce dormancy. Chen et al. [11] and Shahryari et al. [25] reported a high predominance of the *Proteobacteria* group in RO membranes and in osmosed water, respectively. Bereschenko et al. [3] analysed clone libraries of biofilm bacterial communities from different locations of a water treatment plant, detecting that the *Proteobacteria* phylum predominated in all clone libraries and β-proteobacteria class was the most abundant *Proteobacteria* class in tap water and treated water samples, while α-*proteobacteria* was the most abundant in biofilms attached to RO membranes. Therefore, our results do not agree with those of Bereschenko et al. [3], because the PW samples of both centres showed predominance of α-*Proteobacteria*, while in the case of RO membranes, one of the centres showed predominance of α-*Proteobacteria* and the other of β -*Proteobacteria*. One of the variables which may explain the different results of both studies might be the source of the water feed, surface fresh water in the case of Bereschenko and coworkers and potable water in the case of our study. Al-Ashhab et al. [19], by massive sequencing, determined that β-*proteobacteria* followed by γ-*proteobacteria* were the most abundant classes in biofilms formed in RO membranes, which agrees with the metagenomics of DCS RO membrane which showed a higher relative abundance of β-*proteobacteria*. 

Heidarieh et al. [26], studying HD water reported bacterial genera similar to those found in our study. Furthermore, the bacterial genera identified in that study were mainly Gram-negative bacteria, which are commonly related to bacteraemia in patients undergoing HD [16,27]. Bereschenko et al. [3] also reported that most bacteria present in biofilms developed in RO systems are associated to bacteraemia. In our study, *Pseudomonas, Stenotrophomonas* and *Straphylococcus* genera were isolated from RO membranes. These bacterial genera are characterized by forming robust biofilms and being among the main agents associated with bloodstream infections in Chile [28,29,30,31].

*Rastolnia* spp. are considered emerging opportunistic pathogens associated mainly to bloodstream nosocomial infections [32]. We recovered *Rastolnia* sp from RO membranes and osmosed water. This bacterial genus has been characterized as inhabitant of humid environments, including distilled water used for respiratory care at a paediatric intensive care unit and water used for HD, and it was reported in HD patients with bacteraemia [33,34]. Furthermore, *Pigmentiphaga*, characterized for growing in aquatic environments, fresh water and wastewater [29], has been reported as associated to human infections [35] and we recovered *Pigmentiphaga kullae* in this study. To the best of our knowledge, no reports mention the isolation of members of this bacterial genus from water treatment systems in health care environments.

The bacterial diversity at genera level in the two-dialysis centres here studied was heterogeneous and *Pseudomonas* sp. was the bacterial genus most frequently isolated in all samples. There are reports showing that this bacterial genus is one of the main components of RO membrane biofilms, favoured by its high adhesion capacity due to alginate production [20,36]. Alginate is a bacterial exopolysaccharide whose mucous consistency allows bacteria to firmly adhere to surfaces and, in addition, it protects bacteria against antimicrobials, making it an important virulence factor for *Pseudomonas* [37].

The isolation of *Pseudomonas* spp. from biofilm RO membranes is of ecological importance, their exopolysaccharide production favouring biofilm formation and also helping the colonization by other microorganisms, negatively impacting the efficiency of RO membranes. In addition, *Pseudomonas* spp., within Gram-negative bacteria, is the etiological agent most frequently isolated in bloodstream infections of HD patients [38]. *Staphylococcus* spp is another bacterial genus of clinical interest identified, with a high frequency of isolation, in this study. This bacterial genus has also been described as one of the main microorganisms causing infections, including infections in the bloodstream, osteomyelitis and endocarditis, in HD patients [39,40]. Gil et al. [41], studying *S. aureus* biofilm infection in a murine model, showed that exopolysaccharide and extracellular matrix proteins protect bacteria against cells and components of the immune response (neutrophils, macrophages, antibodies and antimicrobial peptides).

It should be noted that the strains isolated from both dialysis centres in this study showed high percentages of resistance to ampicillin, chloramphenicol and cefotaxime. These results are of relevant clinical importance because antibiotics such as beta-lactams and chloramphenicol are the most widely used antimicrobials for the control of human infections, being prescribed in approximately 65% of patients suffering infections [42]. 

Presently, antibiotic resistance, particularly of Gram-negative bacteria, is an emerging problem and the presence of multi-drug resistant bacteria (resistant to at least one antibiotic from three different families of antibiotics [43,44]) in RO membrane biofilms is a major problem. Furthermore, biofilms offer environmental protection and also favour antimicrobial resistance by horizontal transfer of antibiotic resistance genes [45,46,47]. It is also known that microorganisms growing as biofilms develop tolerance to antimicrobials up to a thousand times more than planktonic ones [48]. In addition, subgroups of microorganisms with persistent cell characteristics, such as metabolic inactivity and morphological changes preventing the action of antimicrobials, have been described within biofilms [49,50].

## 5. Conclusions

Although the bacterial composition of biofilms associated to RO membranes depends on physical, chemical and biological factors determined that the bacterial biofilm associated to RO membranes of HD is diverse, with a predominance of the phylum *Proteobacteria*, and *Pseudomonas* sp. was the bacterial genus most frequently isolated. Furthermore, due to its potential virulence and the resistance to antibiotics detected in some of the bacterial species isolated from HD water and RO membranes in our study, they may represent a health risk for patients subjected to haemodialysis treatment.

## Figures and Tables

**Figure 1 microorganisms-08-01418-f001:**
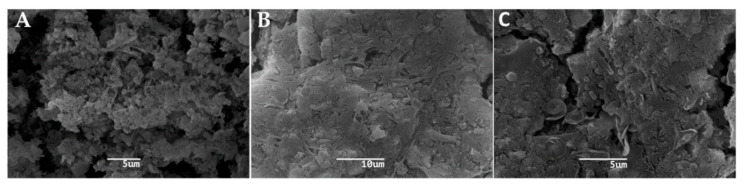
Scanning Electron Microscopy (SEM) of the reverse osmosis (RO) membrane of the north-central area dialysis centre showing the presence of biofilm. (**A**): initial segment of RO membrane. (**B**): middle segment of RO membrane. (**C**): final segment of RO membrane. The scale bar is 5 μm, 10 μm and 5 μm, respectively.

**Figure 2 microorganisms-08-01418-f002:**
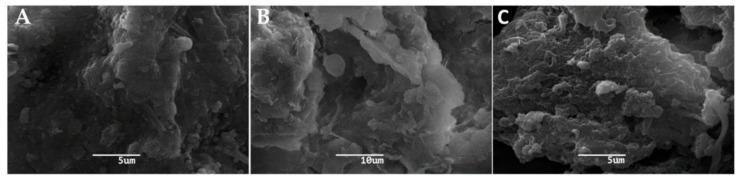
Scanning Electron Microscopy (SEM) of the reverse osmosis (RO) membrane of the south-central area dialysis centre showing the presence of biofilm. (**A**): initial segment of RO membrane. (**B**): middle segment of RO membrane. (**C**): final segment of RO membrane. The scale bar is 5 μm, 10 μm and 5 μm, respectively.

**Figure 3 microorganisms-08-01418-f003:**
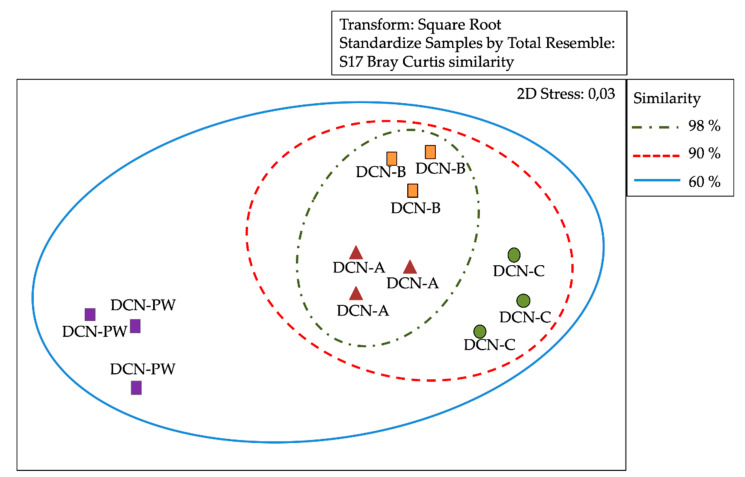
Multidimensional scaling (MDS) of the Denaturing Gradient Gel Electrophoresis (DGGE) data matrix of bacteria 16s rRNA from reverse osmosis (RO) membrane segments and potable water of north-central area dialysis centre (DCN). Similarly, index was evaluated for percentage. DCN-A: initial segment of DCN RO membrane. DCN-B: middle segment of DCN RO membrane. DCN-C: final segment of DCN RO membrane. DCN-PW: DCN potable water.

**Figure 4 microorganisms-08-01418-f004:**
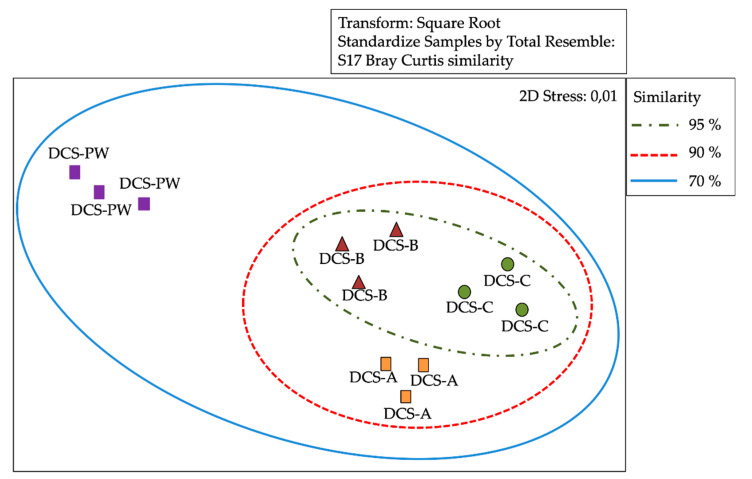
Multidimensional scaling (MDS) of the Denaturing Gradient Gel Electrophoresis (DGGE) data matrix of bacteria 16s rRNA from reverse osmosis (RO) membrane segments and potable water of south-central area dialysis centre (DCS). Similarly, index was evaluated for percentage. DCS-A: initial segment of DCS RO membrane. DCS-B: middle segment DCS RO membrane. DCS-C: final segment of DCS RO membrane. DCS-PW: DCS potable water.

**Figure 5 microorganisms-08-01418-f005:**
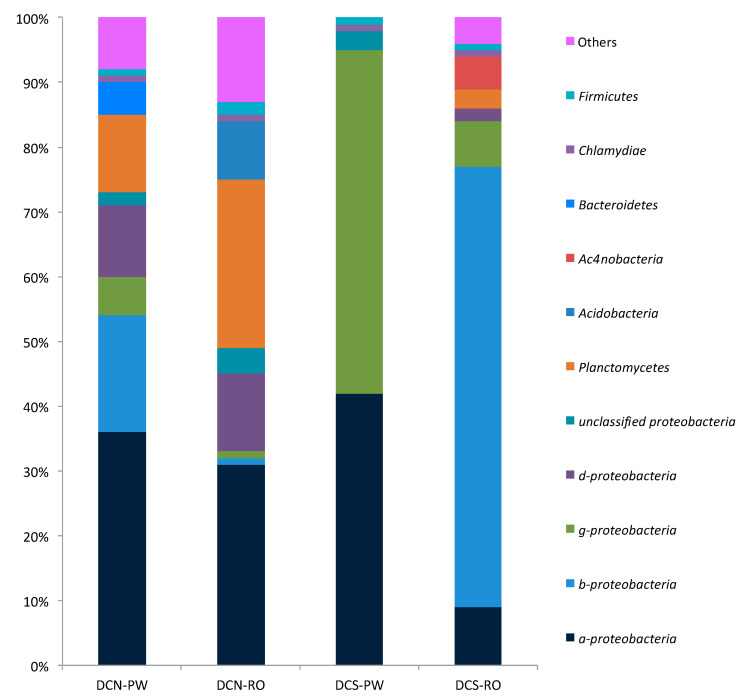
Relative abundance (percentage) of sequences of bacterial phylogenetic groups and *Proteobacteria* subclasses present in potable water (PW) and reverse osmosis (RO) membranes of the north central area dialysis centre (DCN) and south-central area dialysis centre (DCS).

**Table 1 microorganisms-08-01418-t001:** Identification of heterotrophic bacterial strains isolated from both haemodialysis centres.

Strain	Closest Relative Sequence	*Phylum*	Identity (%)	Gen Bank Access Number
DCN RO-3	*Pseudomonas* sp.	*Proteobacteria*	98	MH114031.1
DCN RO-5	*Pseudomonas veronii*	*Proteobacteria*	95	MH482963.1
DCN RO-6	*Stenotrophomonas maltophilia*	*Proteobacteria*	96	FN395264.1
DCN RO-7	*Agrobacterium tumefaciens*	*Proteobacteria*	98	MG575921.1
DCN RO-12	*Pigmentiphaga kullae*	*Proteobacteria*	98	KR856352.1
DCN PW-4	*Ralstonia* sp.	*Proteobacteria*	97	MH844635.1
DCN PW-10	*Arthrobacter* sp.	*Actinobacteria*	95	JQ316230.1
DCN OW-7	*Bacteroides dorei*	*Bacteroidetes*	96	CP008741.1
DCS RO-1	*Pseudomonas* sp.	*Proteobacteria*	98	MK883238.1
DCS RO9	*Staphylococcus* sp.	*Firmicutes*	97	LC435711.1
DCS RO-11	*Staphylococcus pasteuri*	*Firmicutes*	98	MG996881.1
DCS RO-17	*Pseudomonas fluorescens*	*Proteobacteria*	98	EU434450.1
DCS PW-7	*Pseudomonas poae*	*Proteobacteria*	97	MG835964.1
DCS OW-3	*Stenotrophomonas* sp.	*Proteobacteria*	97	KP663381.1
DCS OW-7	*Stenotrophomonas* sp.	*Proteobacteria*	98	KY357351.1

DCN: north-central area dialysis centre; DCS: south-central area dialysis centre; potable water (PW) and of osmosed water (OW); reverse osmosis membrane (RO).

**Table 2 microorganisms-08-01418-t002:** Percentage of antibiotic resistance of the isolated bacterial strains from both haemodialysis centres.

Antibiotics	% Resistant Strains
Ampicillin	69%
Chloramphenicol	55%
Tetracycline	24%
Cefotaxime	53%
Ciprofloxacin	15%
Trimethoprim/sulfamethoxazole	38%
Gentamicin	31%

**Table 3 microorganisms-08-01418-t003:** Antimicrobial susceptibility of bacterial genus isolated from both haemodialysis centres.

Bacterial Genus	AM 10	CHL 30	Te 30	CTX 30	CIP 5	TSX 25	GE 10
*Pseudomonas*	67% (7/11)	55% (6/11)	27% (3/11)	67% (7/11)	18% (2/11)	36% (4/11)	18% (2/11)
*Stenotrophomonas*	100% (4/4)	50% (2/4)	50% (2/4)	50% (2/4)	0% (0/4)	25% (1/4)	50% (2/4)
*Agrobacterium*	100% (0/2)	0% (0/2)	0% (0/2)	100% (2/2)	0% (0/2)	0% (0/2)	50% (1/2)
*Pigmentiphaga*	100% (2/2)	0% (0/2)	50% (1/2)	100% (2/2)	0% (0/2)	0% (0/2)	100% (2/2)
*Ralstonia*	100% (2/2)	50% (1/2)	0% (0/2)	0% (0/2)	0% (0/2)	50% (1/2)	0% (0/2)
*Archobacter*	0% (0/1)	0% (0/1)	0% (0/1)	0% (0/1)	0% (0/1)	0% (0/1)	0% (0/1)
*Bacteroides*	50% (1/2)	0% (0/2)	0% (0/2)	0% (0/2)	0% (0/2)	0% (0/2)	50% (1/2)
*Staphylococcus*	80% (4/5)	40% (2/5)	20% (1/5)	40% (2/5)	0% (0/5)	100% (5/5)	20% (1/5)
** E. coli* ATCC 25922	R	S	S	S	S	S	S
** P. aeruginosa* ATCC 27853	R	R	R	R	S	R	S

* Reference strains: R: resistant; S: sensitive

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
