# Peer review of "Characterization of the Bacterial Biofilm Communities Present in Reverse-Osmosis Water Systems for Haemodialysis"

_microorganisms, 2020, doi:10.3390/microorganisms8091418_

Round 1

Reviewer 1 Report

This is an interesting and clinically useful paper of Cuevas et al which they characterize the bacterial biofilm communities is reverse osmosis (RO), a technique used to treat water systems used for hemodialysis. Further, they authors identified potentially pathogenic bacteria present in the hemodialysis systems of two dialysis centres in Chile.

One of the significances of this paper is that it can help the scientific community to design specific antibiotic treatment against bloodstream infections in patients receiving hemodialysis.

The authors used relevant technologies such as the scanning electron microscopy to demonstrate a heterogenous biofilm which is present on the RO membrane of two hemodialysis centres located in the north-central area (DCN) and south-central area (DCS) of Chile. The authors also test the resistance of bacterial strains isolated from the biofilm on several antibiotics such as ampicillin, chloramphenicol and cefotaxime.

Below are my comments and critiques to the authors:

  1. It will be useful if the authors could mention how many samples they used for these studies and how many replicates they have. It seems that they used 2 samples of the RO membranes for their studies. It will be critical to clearly state that and say whether they have some independent experiments to confirm their observation using the same location water and RO membrane samples.
  2. The authors said that their results are in contradiction from previous reports from Bereschenko el al., a study conducted in Netherlands. Can they authors add one sentence in the manuscript to explain plausible reason of that (locality effect, whether, ….).
  3. The authors used extracellular polymeric substances (EPS) in the introduction and Exopolysaccharide (EPS) in the discussion. Can the authors keep the same abbreviation?
  4. First sentence in the introduction: Add a full stop after “--- to escalate [1]. And please rephrase the whole sentence if possible. “Chronic renal failure is increasing worldwide among all population groups and the incidence of end-stage renal disease (ESRD) continues to escalate [1]. Haemodialysis (HD) is the main treatment for patients with renal failure; therefore, the number of ESRD patients is also increasing worldwide.”
  5. 7. Identification of heterotrophic aerobic bacteria “ --- PCR according to described by Wang et al. (2008) using 16S rDNA” Replace by “ --- PCR according to a method described by Wang et al. (2008) using 16S rDNA”.

Author Response

Dear,

The responses to your review are attached in the attached document.

Regards,

Reviewer 2 Report

Cuevas et al present data regarding the bacterial composition of water and membranes in dialysis reverse osmosis systems from two centers in Chile. Bacteria are characterized from the potable water supply, three sections of the reverse osmosis membrane, and the osmosed water. They use scanning electron microscopy to identify biofilms formed on the RO membrane sections, and perform both total and heterotrophic bacterial counts from each sample. To characterize bacterial composition, they performed 16S rRNA PCR followed by DGGE and Illumina sequencing. Furthermore, for the bacteria that they were able to culture, they performed specific identification of isolates and antimicrobial susceptibility assay. They find that the most commonly isolated phylum isolated was proteobacteria, with Pseudomonas sp. making up a large share of biofilm formation. Antimicrobial susceptibility testing highlighted a large degree of resistance amongst the isolates obtained, which has important implications for patients.

Overall, the manuscript is logically structured and well written. The experimental approach is well thought out and thorough with respect to the questions asked. There are, however, a few items that should be addressed prior to publication.

  • Lack of control data: It would be nice to show what a clean RO filter looks like under SEM for comparison in Figures 1 and 2. What negative controls were used for bacterial counting experiments to rule out contamination? Also, the results from antimicrobial susceptibility testing on the reference strains mentioned in the methods is not shown.
  • Regarding bacterial counts: As it reads, there were consistently higher bacterial counts in the osmosed water compared to the potable water (although it is stated that these were not statistically significant differences) – this is surprising, given the purpose of reverse osmosis. If the data given is an average, it might be useful to provide the raw data in table form or at least +/- SD values to give the reader an idea of the variability observed.
  • If indeed there were more bacteria in OW, why were you unable to extract sufficient DNA for sequencing experiments? Was this due to the different extraction kit used for sequencing versus PCR-DGGE?
  • Why were the PCR-DGGE results not shown for OW in Figures 3 and 4?
  • For Figure 5, it would be useful to rearrange the bars on the graph such that they can be more easily compared: i.e: DCN-PW, DCN-RO then DCS-PW, DCS-RO. I would have assumed that the bacterial composition of the RO membrane would represent a sub-sample of bacteria from the PW, so it is surprising that the DCS-RO composition is so wildly different from the DCS-PW, particularly with regards to the b-proteobacteria, which make up the majority of the DCS-RO but appear absent in the DCS-PW.
  • Minor typos: please double check that acronyms used are consistent defined at first use. For example, “DGGE” and “OTU” are not defined. There is also some inconsistent use of “WO” versus “OW” and “WP” versus “PW”. Particularly in section 3.2.

Author Response

(The authors gave the same response as above.)
